# HIGHLY EFFICIENT SPEECH SEPARATION USING RELATIVE CONTEXT

## ABSTRACT

Speech separation is a problem area where a mixture with overlapping speech signals is the input and estimations of the clean speech signals which make up the mixture is the output. In this paper we propose a novel sequence modelling method called relative context and use it for a speech separation architecture called RCSep.

The main advantages of relative context is that it does not require trainable parameters, is very lightweight and highly parallelized. The RCSep model which heavily uses relative context is an extremely efficient source separation model. It has less than 500k trainable parameters, lower memory usage and is significantly faster than all previous source separation methods while still maintaining high separation accuracy.

Furthermore, we also used relative context instead of LSTMs in a current SOTA architecture which simultaneously improved separation accuracy and decreased computation time, memory usage and model size.

## 1 INTRODUCTION

### 1.1 BACKGROUND

Audio source separation is a signal processing problem which in the last decade has seen major advancements using machine learning. The aim of audio source separation is to recover the individual sources that make up a mixture given only the mixture. For example, when multiple people are talking over each other, they create a mixture and the goal of a source separation system is to estimate the original utterances of each speaker. This problem is also known as the cocktail party problem (Bronkhorst, 2000; Haykin & Chen, 2005).

Expressed formally, the mixture $\vec{x} \in \mathbb{R}^{L \times 1}$ is the sum of the $C$ individual audio signals $\vec{s_1} \in \mathbb{R}^{L \times 1}$ to $\vec{s_C} \in \mathbb{R}^{L \times 1}$

$$\vec{x} = \sum_{i=1}^{C} \vec{s_i} \tag{1}$$

with $L$ being the sequence length and $C$ being the number of individual sources which the separation system is trying to recover. In the context of this paper, we focus on single-channel source separation. Single-channel simply means that the audio was recorded using a single microphone. This area of research is relevant to any other problem which struggles with noisy inputs due to overlapping signals (Narayanan & Wang, 2014). Some notable examples include automatic speech recognition (ASR), music and audio production and hearing devices.

### 1.2 MOTIVATION

In the last few years, research for single-channel speech separation has been advancing quickly. In the Conv-TasNet paper (Luo & Mesgarani, 2019), people were asked to rate the estimations the model produced against the clean baseline on a scale of 1 to 5 with 5 being the best quality. The estimations of the Conv-TasNet almost matched the results of the clean signal with the estimations

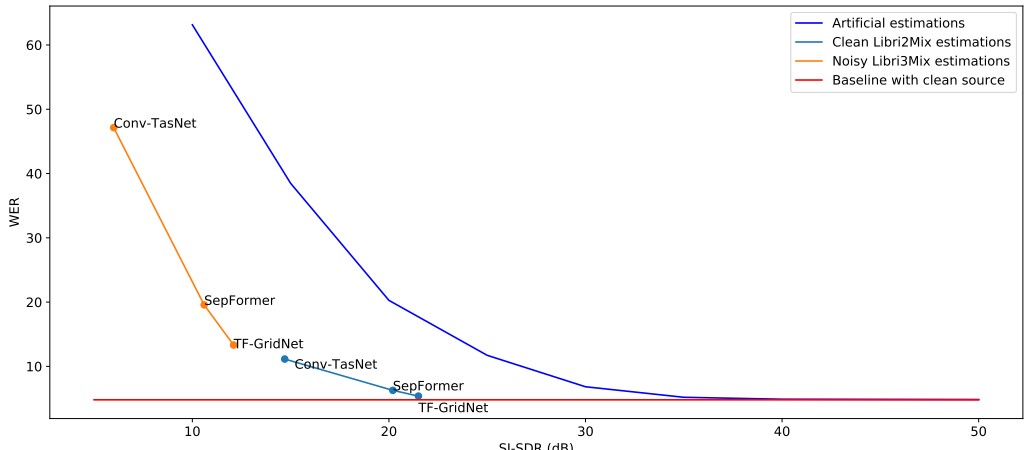

Figure 1: WER of LibriSpeech test-clean ASR benchmark using wav2vec2 for ASR in relation to the SI-SDR of the input audio. Artificial estimations were mixed together at the specified SI-SDRs while the other estimations are a results of the specific separation system being applied to the clean Libri2Mix and noisy Libri3Mix. Once the estimations reach the same WER as the baseline clean source, further separation accuracy improvement is unnecessary for ASR.

reaching a mean score of roughly 4 while the clean signals reached a score of 4.25. For human listening, the Conv-TasNet estimations are already almost as good as clean signals.

Since this is a very subjective measure, however, and speech separation can be used as a preprocessing step for other tasks, we did some additional testing. The goal is to find out, whether separation accuracy improvements are still relevant to other tasks or whether the separation accuracy has already passed a certain threshold where further improvement is practically irrelevant. In order to test this, we ran an experiment using the LibriSpeech Corpus (Panayotov et al., 2015) and two speech separation datasets which are based on this corpus (Cosentino et al., 2020). We test the ASR performance of the wav2vec2 (Baevski et al., 2020) model using clean sources from the LibriSpeech corpus test-clean set as the baseline level and various estimations of the sources which are produced by a source separation model. The intent is to find out whether these estimations of the sources can reach the baseline level, which would mean that further separation accuracy improvements are irrelevant for ASR, at least for the wav2vec2 model.

We show the results of this experiment in Figure 1. Figure 1 shows the performance of the wav2vec2 model in word error rate (WER) compared to separation accuracy in scale-invariant signal-to-distortion ratio (SI-SDR) (Roux et al., 2018). The baseline of this model is 4.8% WER for clean audio input as depicted in the red line. This is slightly worse than the results reported in the original paper because the original paper uses a sampling rate of 16 kHz while we use 8 kHz which is then upsampled. The reason we do this is because in audio separation it is currently common practice to work with 8 kHz data and all the pretrained models we use were trained with 8 kHz data.

The dark blue line shows artificially mixed together audio signals at the given SI-SDRs. For these artificial estimations, the point at which the baseline WER performance is reached, is about 35 dB SI-SDR. However, these artificial estimations mix together two speech signals at a constant rate, while real separation models do not operate like that. Therefore, we also show experiments using models trained on two speaker separation data with no background noise (light blue line) and three speaker separation data with background noise (orange line).

Our experiments show that although these tasks have different difficulties and therefore different results for the models, they seem to follow a fairly consistent pattern. For both datasets we use the same three models: Conv-TasNet (Luo & Mesgarani, 2019), SepFormer (Subakan et al., 2021) and TF-GridNet (Wang et al., 2022). The results of this experiment show, that for the easier two speaker separation task, current SOTA models like TF-GridNet produce estimations that reach almost 5% WER which makes them basically equivalent to the clean sources at 4.8% WER.

As the current SOTA speech separation models for two speaker separation without background noise have already reached the threshold at which further separation accuracy improvement is irrelevant, the logical next step is to find more lightweight solutions which can approach this threshold. Currently, most speech separation models use millions of parameters and are difficult to run in real-time, especially on low resource devices like hearing devices. Therefore, finding an efficient and accurate speech separation model is the topic of this paper.

## 1.3 CONTRIBUTIONS

This paper has the following three contributions:

1. We introduce the relative context operation which allows for pattern recognition within neural networks without using trainable parameters.

2. Using the relative context operation, the RCSep architecture is constructed which performs single-channel speech separation with high accuracy, very few trainable parameters, high speed and low memory usage. To our best knowledge, the RCSep outperforms all previous separation models in training speed, inference speed, and training memory usage while matching the previous bests for inference memory usage. In terms of model size, the RCSep is over 3 times smaller than previous lightweight models while maintaining comparable accuracy on the WSJ0-2Mix and WHAM! benchmark.

3. We determine a threshold value for source separation at roughly 25 dB SI-SDR at which further accuracy improvements are irrelevant for ASR and likely most other tasks.

## 2 RELATED WORKS

As with most problem areas where pattern recognition is necessary, modern source separation systems rely on neural networks. In early deep learning research concerning source separation, the separation approaches usually were based on the short-time Fourier transform (STFT) (Hershey et al., 2016; Kolbaek et al., 2017; Luo et al., 2018). The magnitude information of the mixture was used as the input of the neural network and corrected magnitudes for each estimation were calculated as the output. These new magnitudes alongside the mixture's phase information were then used to return to waveforms using the inverse STFT. This approach, however, was limited by not changing the phase information. The reason why changing the phase information is not as straightforward as correcting the magnitude (Williamson et al., 2016) is due to the phase being the imaginary part of the complex valued STFT while the magnitude is the real part.

In order to remove this upper limit set by not changing phase information, time domain systems were proposed instead, initially in (Wang & Wang, 2015) and later in (Luo & Mesgarani, 2018) which set the foundation of current time domain source separation systems. The main advantage of time domain based separation approaches was that it would not decouple phase and magnitude information and just operate on the waveform directly instead.

Further improvements to the time domain based approaches include the dual-path method (Luo et al., 2020a) as well as the use of Transformers (Vaswani et al., 2017) within the context of source separation (Chen et al., 2020; Subakan et al., 2021). The main idea of the dual-path approach is to split the input mixture into overlapping chunks and then stack these chunks on top of each other. The neural network uses layers which are capable of capturing sequential patterns across the sequence inside the chunks (intra-processing) as well as the sequence of the chunks (inter-processing). This allows for local and global pattern recognition and generally resulted in higher separation accuracy (Luo et al., 2020a; Chen et al., 2020; Subakan et al., 2021; Lam et al., 2021; Rixen & Renz, 2022b). In some more recent research, frequency domain separation methods (Wang et al., 2022; Yang et al., 2022) have been competitive with the best time domain methods (Rixen & Renz, 2022a; Jiang et al., 2024; Zhao et al., 2023; Lee et al., 2024; Mu et al., 2023; Yip et al., 2024) as working with complex valued tensors is now supported in most deep learning frameworks. There have also been some models which combine time- and frequency domain approaches (Rixen & Renz, 2022b; Lutati et al., 2023) and reach SOTA performance.

However, all these methods are approaching the threshold value determined in Figure 1 which is why finding more efficient models is becoming more relevant. Notable lightweight separation methods include the group communication method (Luo et al., 2020b) and small versions of certain models

like S4M (Chen et al., 2023). There are some other relevant methods like the TDANet (Li et al., 2023), tiny SepFormer (Luo et al., 2022), small version of DP-Mamba (Jiang et al., 2024) and the Sandglasset (Lam et al., 2021) which do have a focus on efficiency, but as they still exceed 2 million trainable parameters they cannot be considered as lightweight as the previously mentioned models. In fact, even the small version of S4M has almost 2 million trainable parameters, meaning there is a severe lack of research for tiny models, with the only real exception being the group communication paper. While the group communication method is extremely effective at lowering model size, other efficiency metrics like speed and memory are still problematic and often even negatively effected by using this method.

To summarize, current lightweight separation methods are not exceeding in every efficiency metric. Some focus on memory usage (Lam et al., 2021) and speed (Li et al., 2023; Chen et al., 2023), others on model size (Luo et al., 2020b). In this paper, we propose the RCSep architecture which to our best knowledge outperforms all previous methods in terms of speech and memory usage. The model size of the RCSep is significantly smaller than all previous models at less than 500k trainable parameters except for the group communication method. Separation accuracy of the RCSep is also improved in comparison to the previous lightweight models and even borderline reaches SOTA.

RCSep draws inspiration from some previous work, specifically the Sandglasset (Lam et al., 2021), QDPN (Rixen & Renz, 2022a) and some of the hybrid models combining time- and frequency domain approaches (Lutati et al., 2023; Rixen & Renz, 2022b). We did also test out the group communication method in order to lower model size even further, however this lead to lower separation accuracy and increases in memory usage and computation time. There are, of course some other proven methods for lowering model size and computational cost like weight sharing and quantization, however, these methods also tend to have a negative impact on accuracy. Since the model size of the RCSep is already extremely small at less than 500k parameters, we instead elected to keep its accuracy higher.

## 3 RELATIVE CONTEXT

The basic idea of relative context is to shift the input tensor across a given axis where patterns exist (e.g. height and width for images, time for audio, etc.) and subtract it from the original input tensor. Instead of describing the input with raw values, relative context describes them as offsets in relation to previous or following elements. This makes relative context a type of differencing operation.

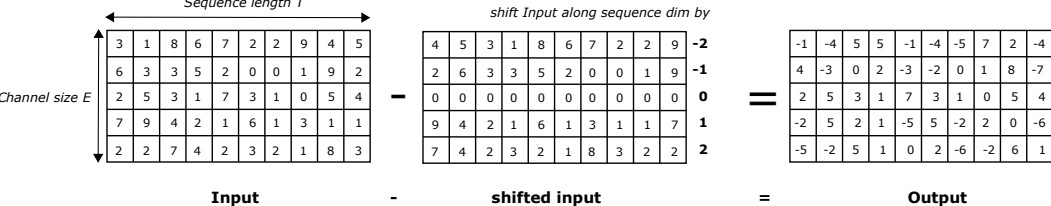

Figure 2: The relative context operation across one dimension with $K$=5. For the subchannel where the shift is 0, no subtraction happens since it would result in deleting the information of that subchannel and the input is instead preserved.

There are many input types where this idea is useful. For images, the raw values just describe how bright the pixel is in the channel. If we apply relative context, however, we can instead get to know how bright this pixel is in relation to its neighbours.

Generally speaking, most neural network architectures use a channel size that is much greater than that of the original data input. The relative context operation makes use of this fact and splits its input across the channel dimension into $K$ subchannels. Each subchannel is then shifted by a different amount. An example of this is shown in Figure 2 where both the channel size $E$ and the number of subchannels $K$ are set to 5. The input tensor is split into 5 subchannels with the first row being shifted two elements to the right and the last row being shifted two elements to the left. This shifted

tensor is then subtracted from the original to produce the output of the relative context operation. Note, that for the subchannel where the shift is equal to 0, we simply copy the original input into the output. Otherwise, we would delete information and have a row of zeros.

The relative context operation is named after the idea that it delivers information about the current element in relation to its neighbours. It enables sequence modelling without adding trainable parameters which is why it is very effective for lightweight models. One disadvantage of the relative context operation is that it links the channel size to the amounts of shifts that are possible. At most, one can set $K = E$ for the maximum amount of shifts. However, this would result in subchannels of size 1. In our experiments, this never lead to optimal accuracy since some shifts are more important than others, specifically the smaller ones which give context in relation to the direct neighbours. Therefore, leaving them a greater subchannel size by keeping $K$ relatively low usually is the better choice for achieving greater accuracy.

The relative context operation in its current form basically encourages the neural network to place relevant information for the specific shifts into their corresponding channels.

Relative context can be applied across a single dimension for one dimensional data like audio or across multiple dimensions for data like images. Figure 2 shows a simple example of applying one dimensional relative context. Both the Figure 2 and the rest of the paper assume relative context to be bidirectional, however, it is easily possible to make it unidirectional and have it work for real-time applications.

Since the way relative context works is somewhat similar to convolutional layers, we also include the option of setting a dilation factor to enable a stack of relative contexts to behave like temporal convolutional networks (TCN) (Lea et al., 2016). The inclusion of the dilation factor is very straightforward, as one just multiplies the shift by the dilation factor to get the new shift.

## 4    RCSEP

The RCSep model uses a hybrid approach where a time domain model produces the initial estimations which are then used alongside the input mixture for the frequency domain model to output the final estimations. An overview of the RCSep architecture is shown in Figure 3.

### 4.1    TIME MODEL

The time model produces the first set of estimations. As the name suggests, it is a time domain based model. The initial estimations and the original mixture are later used for the frequency model to produce the final estimations.

#### 4.1.1    ENCODER

Similar to what was done in the Sandglasset and the QDPN model, we first segment the input mixture into overlapping chunks with an overlap ratio of 50%. This temporarily doubles the tensor size, however, as we use chunk size $M$ as our channel dimension, we effectively halve the sequence size $L$ through this step, massively improving computational cost. In our testing, this step also slightly increases accuracy. After the chunking step, the tensor is fed through a one dimensional convolutional layer which increases the channel size from $M$ to $E_T$ with $E_T$ being the channel size of the time model. The kernel size and stride of the encoder are set to 1.

#### 4.1.2    SEPARATION

The general structure of the separation module is also inspired by the Sandglasset and the QDPN. Similar to the QDPN, we combine a TCN and Transformer architecture. The first difference is, that the RCSep uses relative context as is shown in the purple box in Figure 3 instead of convolutional layers which massively decreases computational cost and model size. The structure of the temporal relative context network (TRCN) is shown in the purple box in Figure 3. For the time model, a depth of 8 layers is chosen where the dilation factor increases from $2^0$ in the first layer to $2^7$ in the last layer. The other difference is the usage of what we call MiniFormer blocks. Just like in the QDPN and Sandglasset, depthwise convolutional layers are used for downsampling and upsampling before

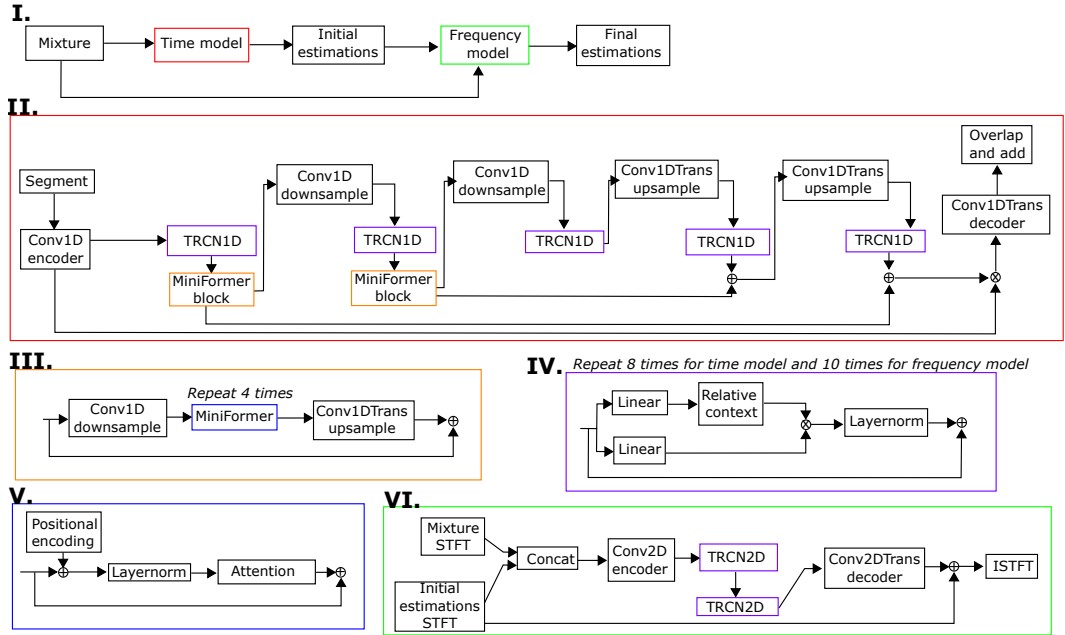

Figure 3: The RCSep architecture. Details on what each step contains is shown in the color matching boxes. I. Overview of the RCSep architecture. II. Overview of the time model. III. Structure of the MiniFormer block. IV. A single relative context block which forms the temporal relative context network (TRCN) by being repeated 8 times for the time model and 10 times for the frequency model. V. Structure of a single MiniFormer which is repeated 4 times for each MiniFormer block. VI. Overview of the frequency model.

and after 4 MiniFormers as shown in the orange box in Figure 3. The MiniFormer structure is shown in the dark blue box in Figure 3. Unlike normal Transformers, it does not include the feed forward network to save on computational cost. The attention layer also has a slight adjustment which in our testing did not affect accuracy while lowering model size and computational cost. Unlike a normal attention layer which uses multiple linear layers for the query, key and value, in our implementation they are just multiplied by three weights, each with a size of $E_T$.

Note, that the entire separation module does not change the channel size at any point. The only axis which does change is the sequence axis. This is inspired by the Sandglasset which in turn is based on the many successful applications of the U-net (Ronneberger et al., 2015). After each of the first two TRCNs, a downsampling step is performed using depthwise convolutional layers. The third TRCN is equivalent to the bottleneck layer, after which upsampling occurs using depthwise transposed convolutional layers. As is shown in Figure 3, residual connections between the tensors with the same sequence size exist, meaning the result of the fourth TRCN is added to the result of the second TRCN and the same happens for the first and last TRCN. In our testing this U-net structure slightly increased accuracy while significantly lowering computational cost.

The last step of the separation module is to multiply its output with the encoded input from the beginning of the network.

### 4.1.3 DECODER

The decoder is simply a one dimensional transposed convolutional layer which changes the channel size to $M \cdot E_T$ and allows for the $C$ estimations to get reconstructed into a waveform using the overlap and add operation with the same parameters as the initial segmentation that occurred before the encoding step.

Note, that for the time model to actually produce estimations, this output needs to be used for the loss calculation in addition to the final output.

## 4.2 Frequency model

The frequency model takes both the original mixture and the $C$ estimations produced by the time model as its input. Generally speaking, the goal of the frequency model is to correct these initial estimations. The time model is responsible for most of the work and most of the computation. We did, however, find that adding this frequency model was more effective for increasing accuracy than using a bigger time model.

### 4.2.1 Encoder

The first step of the frequency model is to apply the STFT to the mixture and the estimations and then concatenate them together. Specifically, the tensor shape for the frequency model is four dimensional unlike the time model which was three dimensional. These four dimensions are the batch dimension, the time dimension of the bins, the frequency dimension of the bins and finally the channel dimension. Note, that since the STFT outputs complex numbers, the channel dimension for each STFT has a size of 2, storing the real and imaginary values of the STFT. The STFTs are concatenated in said channel dimension and fed through a two dimensional convolutional layer which expands the channel size to $E_F$ which is the channel size of the frequency model.

### 4.2.2 Separation

As previously mentioned, the frequency model is much smaller than the time model. The separation module, which makes up the bulk of the computation for either model, only consists of two TRCNs. The difference for the frequency model is that said TRCNs are applying two dimensional relative context across both the time and frequency axes of the bins and that the TRCNs have a depth of 10 instead of 8. In this case the dilation factor increases from $2^0$ to $2^9$ from the first to the last layer. In our experiments, adding MiniFormer blocks or a U-net structure did not improve accuracy which is why we elected to only use TRCNs.

### 4.2.3 Decoder

The decoder is a transposed two dimensional convolutional layer which returns the channel size from $E_F$ to $2 \cdot C$ since we require two channels for each estimation for both the real and imaginary parts of the STFT. These corrections to the estimations STFTs are then added to the original STFTs before going through the inverse STFT to reconstruct the estimations as waveforms.

## 5 Experiments

### 5.1 Datasets

We evaluated the RCSep model on two speech separation benchmarks, the WSJ0-2Mix (Hershey et al., 2016) and the WHAM! dataset (Wichern et al., 2019). Both datasets are based on the WSJ0 corpus (Garofolo, John S. et al., 1993). The WSJ0-2Mix is a two speaker separation dataset without background noise and without reverberation while the WHAM! dataset includes background noise.

Each dataset contains 30 hours of training, 10 hours of validation and 5 hours of evaluation data. 119 different speakers with roughly half being female and the other half being male are included. Different utterances but the 101 same speakers are used for the training and validation sets while the evaluation set has both different utterances and 18 different speakers than the training and validation sets.

### 5.2 Model configuration

The chunk size for the segmentation, $M$, is equal to 4 for all experiments. This is optimal for accuracy, but it is possible to lower computational cost further by increasing this value since it would lower the sequence length. The channel size of the time model $E_T$ is set to 64. This means

that the group sizes of all down- and upsampling layers is also equal to 64 since they are depthwise convolutional layers. The stride factor of the two downsampling layers in the U-net structure is set to 2 and 8, respectively. The kernel sizes of these convolutional layers is double that of their stride factor. The kernel sizes and stride factors of the transposed convolutional upsampling layers are the same as the corresponding downsampling layers.

The MiniFormer blocks use down- and upsampling layers with a stride factor of 32 and a kernel size of 64. The group size is once again equal to the channel dimension, making them depthwise convolutional layers. The number of subchannels $K$ of the relative context operations is set to 7 for the time model and 3 for the frequency model.

For the STFT, the window size is set to 256 while the hop size is set to 64. While we have found this setup to be optimal for accuracy, it is possible to significantly lower computation time of the RCSep by lowering the window size to 128. This does lower accuracy a bit, but since it also halves the tensor size for the frequency model, it has a significant impact on both speed and memory usage. Since the RCSep is already outperforming all previous models in these metrics, however, we elected to prioritize accuracy but for actual deployment it might make sense to lower the window size to 128. The channel size of the frequency model, $E_F$, is 64.

We train the RCSep for a total of 200 epochs. For the first 100 epochs, we use a learning rate of $1e-3$ and after the 100th epoch we halve the learning rate if the validation SI-SDR does not improve for 3 consecutive epochs. Gradient clipping with a maximum L2 norm of 5 is employed in order to avoid the exploding gradient problem. The Adam optimizer (Kingma & Ba, 2017) is used.

We use the standard loss function for speech separation, meaning the SI-SDR. Since we have two sets of estimations, however, we also need to calculate two losses which are then summed up for a final loss before the backwards pass.

Aside from the RCSep architecture, we also tested a TF-GridNet variant, where we replace the BLSTMs with one dimensional TCRNs with a depth of 7.

## 5.3 RESULTS ON WSJ0-2MIX AND WHAM!

We show the results of our experiments in Table 1. We include two versions of the RCSep model, with the RCSep128 having double the channel size in the time model compared to the RCSep64.

Table 1: Comparing the model size and scale-invariant signal-to-distortion ratio improvement (SI-SDRi) on the WSJ0-2Mix and WHAM! of previous models and our proposed model, the RCSep. We include two versions, the RCSep64 with a channel size of 64 for the time model and RCSep128 which has a channel size of 128 for the time model.

| Method | Model type | Model size | SI-SDRi (dB) | |
| --- | --- | --- | --- | --- |
| | | | WSJ0-2Mix (Hershey et al., 2016) | WHAM! (Wichern et al., 2019) |
| Conv-TasNet (Luo & Mesgarani, 2019) | Time | 14.9M | 15.3 | 12.7 |
| DualPathRNN (Luo et al., 2020a) | Time | 2.6M | 18.8 | 13.7 |
| TDANet (Li et al., 2023; Chen et al., 2023) | Time | 2.3M | 18.6 | 15.2 |
| S4M-tiny (Chen et al., 2023) | Time | 1.8M | 19.4 | - |
| SepFormer (Subakan et al., 2023) | Time | 26.0M | 22.3 | 16.4 |
| TF-GridNet (Wang et al., 2023) | Frequency | 14.5M | 23.5 | - |
| MossFormer2 (Wang et al., 2023) | Time | 55.7M | **24.1** | **18.1** |
| RCSep64 | Hybrid | **485K** | 17.8 | 13.4 |
| RCSep128 | Hybrid | 1.38M | 19.4 | 14.8 |
| TF-GridNet + TRCN | Frequency | 12.1M | 23.7 | - |

While the RCSep is unable to match current SOTA models like TF-GridNet and MossFormer2, these models are over 10 times bigger and have a significantly higher computational cost. Therefore, they are not really in direct comparison with the RCSep models. The methods that make a more fair comparison are the recent lightweight separation methods such as the TDANet and S4M-tiny. Note, that both RCSep models are still significantly smaller than the TDANet and the S4M-tiny. While the RCSep64 is not quite able to match their accuracy on the WSJ0-2Mix and WHAM! benchmarks, it still is fairly close, reaching 17.8 dB on the WSJ0-2Mix and 13.4 dB on the WHAM! dataset. The RCSep128, however, is able to match and even outperform the S4M-tiny and TDANet in terms

of separation accuracy, reaching an SI-SDRi of 19.4 dB on the WSJ0-2Mix and 14.8 dB on the WHAM!.

The TF-GridNet variant which uses TRCNs instead of BLSTMs reaches and SI-SDRi of 23.7 dB, which is marginally higher than the original's 23.5 dB. It is, however, also significantly smaller than the original at 12.1 million trainable parameters instead of 14.5 million trainable parameters.

## 5.4 RESULTS COMPUTATION TIME

Figure 4 shows the computational cost in terms of training and inference speed. We compare the two RCSep models with two recent lightweight models, the TDANet and S4M-tiny, as well as some larger models like SepFormer and TF-GridNet plus the TF-GridNet variant with TRCNs. We use an input with a sequence length of 32000 and do a 1000 runs for the speed tests.

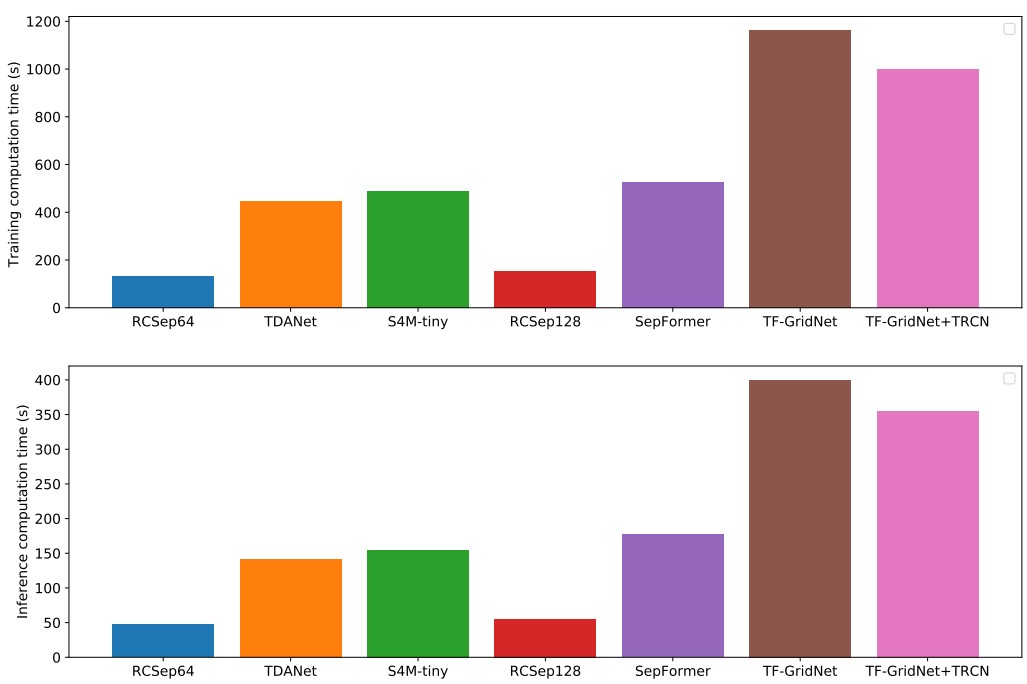

Figure 4: Training and inference computation time from various speech separation models given a 4 second 8 kHz input measured by using 1000 runs. The models are listed in order of separation accuracy, with the least accurate models starting from the left.

For the speed tests, both RCSep models significantly outperform not only the bigger models like the SepFormer and TF-GridNet but also the recent lightweight models, meaning the TDANet and S4M-tiny. The RCSep128 model is only very slightly slower than the RCSep64 model despite being significantly more accurate. Both models are roughly 3 times faster than any of the other models we tested during both training and inference.

Furthermore, the TF-GridNet variant using TRCNs is 14% faster than the original during training and 12% faster during inference while also being more accurate and having 17% fewer trainable parameters.

## 5.5 RESULTS MEMORY USAGE

Figure 5 shows the memory usage of the same models as 4 during training and inference while processing a 4 second 8kHz input. Memory usage during inference is fairly uniform across all models tested except for the original TF-GridNet which uses about twice as much memory as all

the other models. The TRCN version of the TF-GridNet reduces inference memory usage by 45%. Training memory usage between these two models, however, is basically identical. The RCSep models are in line with the other lightweight models during inference, but use roughly 2-3 times less memory than any of the other models during training. The RCSep128 uses slightly more memory than the RCSep64 during both training and inference but for most application this would likely be a worthwhile trade off considering the accuracy difference between the two models.

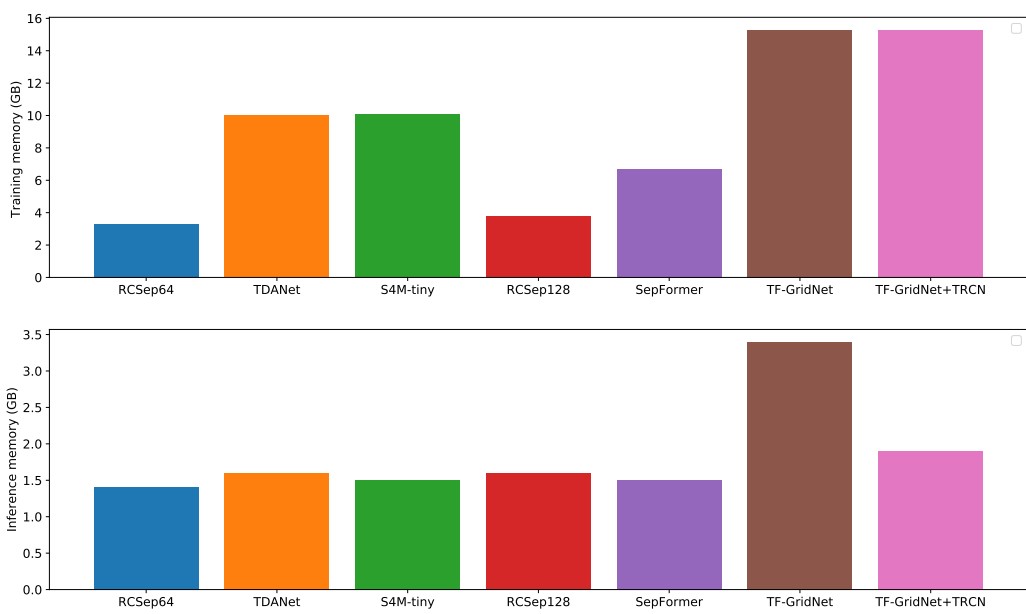

Figure 5: Training and inference memory usage from various speech separation models given a 4 second 8 kHz input. The models are listed in order of separation accuracy, with the least accurate models starting from the left.

## 6 CONCLUSION

In this paper we introduced the relative context operation as well as the RCSep architecture for speech separation.

Through the experimental results shown, it is clear that the RCSep has by far the lowest computational cost of any speech separation architecture while also matching or even outperforming previous lightweight models in terms of separation accuracy.

We have further demonstrated the potential of the relative context operation by using it in a current SOTA model, the TF-GridNet, instead of BLSTMs. This resulted in a slight accuracy gain, a 17% reduction in model size, a 10-15% speed increase and a 45% memory usage decrease during inference. Therefore, using relative context instead of BLSTMs caused a significant drop in computational cost while marginally increasing the separation accuracy.

Additionally, we have experimentally determined a threshold value of roughly 25 dB SI-SDR at which further improvement is irrelevant to separation accuracy. This means, that making separation models more lightweight is the next most important task in this problem area. While the RCSep models are still relatively far from this threshold, the modified TF-GridNet model does almost reach it while being significantly more lightweight than the original. Furthermore, for other applications such as human listening, the RCSep models already produce higher quality estimations than the Conv-TasNet model whose estimations were rated almost on par with the original sources. We will provide audio samples at a later date.

## 7  REPRODUCIBILITY STATEMENT

The datasets used are described in section 5.1. No special preprocessing is used. The relative context operation itself is described in section 3 and shown in Figure 2. The RCSep architecture is described in section 4 and shown in Figure 3 while its parameters are defined in section 5.2.

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
