# OpenReview forum: "Highly efficient Speech Separation using relative Context"
_ICLR.cc/2025/Conference — Submitted to ICLR 2025_

### Official Review · Reviewer_ei5R · 2024-10-30

**Soundness:** 2
**Presentation:** 1
**Contribution:** 3
**Rating:** 3
**Confidence:** 4

**Summary:**

The paper describes a lightweight model for speech separation. It starts with describing the results of an investigation into the benefits of further improving the separation quality, which is performed using a subsequent ASR task. It concludes that at least for the ASR downstream task, there is not much to gain from further improving the separation quality, and therefore, the most important point is to reduce model complexity, without degrading performance.

Subsequently, the paper introduces the RCSep model, which includes a new relative context operation that is presented as the main novelty of the paper.  The paper briefly describes the model architecture, which is a hybrid architecture working in both, the time and frequency domains.

The experimental evaluation compares the model with quite a few state-of-the-art models of various complexities, to three proposed models: two variants of the RCSep model and a modification of the TF-Gridnet which was modified to include the proposed temporal relative context networks (TRCN) instead of the BLSTM.

Overall, the performance of the proposed RCSep model is far below what is possible with today's state-of-the-art models. The RCsep model compares favorably with other lightweight models though. Interestingly, the TF-GridNet using the TRCN network achieves slightly better performance, than the original version that uses BLSTM layers.

**Strengths:**

### Strengths

- the initial comparison concerning the benefits of further improving the separation quality based on studying the impact of the separation quality on a downstream ASR task,
- potentially interesting idea,
- good results notably for the TF GridNet with TRCN layers.

**Weaknesses:**

### The initial conclusion about the need for further improvements in separation quality appears not fully justified

The initial experiment on the benefits of further improving separation quality considers 2 speakers without background noise. In nearly all practical applications there is background noise though, and therefore it appears to me that the conclusion of the initial discussion would require more caution.


### The presentation is quite difficult to follow and contains technical errors

**Authors write on line 140:** *The reason why changing the phase information is not as straightforward as correcting the magnitude (Williamson et al., 2016) is due to the phase being the imaginary part of the complex valued STFT while the magnitude is the real part.*

Comment: First for becoming technically correct the term "log" needs to be put somewhere. Magnitude and phase are the real and imaginary part of the logarithm of the complex DFT spectrum. But even when the phrase is fixed, I don't see any reason why being the real or imaginary part of a complex representation would have an impact on the difficulties encountered during separation. I could not find such a statement in Williamson 2016 either.

### Wrong claim

**Authors write on line 160**: *However, all these methods are approaching the threshold value determined in Figure 1 which is why
finding more efficient models is becoming more relevant.*

As mentioned above this is only true for the practically less relevant task of two speakers without any background noise.

### Notation, terms, and explanations are confusing

- Most of the constants (K, E, M, ..) are introduced on the fly. It would have been helpful to introduce them in the beginning around eq 1, where we find only L and C.

- What is denoted a *shifting operation* in Fig 2, appears to be a *circular shift*.

- The main novelty, the "relative context operation", has not been explained in a way that allows easily understanding what is happening.

**Authors write in line 212**: *Generally speaking, most neural network architectures use a channel size that is much greater than
that of the original data input. The relative context operation makes use of this fact and splits its input across the channel dimension into K subchannels. ...*

A few words are missing here, it could be read as: the method splits the channels "into groups with K sub-channels". One could also assume that it was meant that there should be K groups with S subchannels. Given this ambiguity, it would have been much clearer if in figure 2 authors would not haven chosen K=E.

**Line 213**: *Each subchannel is then shifted by a different amount. An example of this is shown in Figure 2 where both the channel size E and the number of subchannels K are set to 5.*

If we have K=E sub-groups have size 1 and one channel can not be shifted against itself. So here it becomes clear that the confusion comes from the fact that authors denote as subchannels what is a group of sub-channels, and that the group of subchannels are not shifted against each other but shifted synchronously all together against all the other groups.

### Experimental evaluation is insufficient

It would have been interesting to make a few ablation studies K=1, K=10, ... K=E.

### Discussions

The paper does not provide any discussion about why this method is working comparatively well with comparatively few parameters.

**Questions:**

**paper title**

why do you name this method "relative context". I understand your method as an alternative implementation of a Conv operator, where a few weights are fixed to zero.


**You write in line 223**: *At most, one can set K = E for the maximum amount of shifts. However, this would result in subchannels of size 1.*

Can you explain why this (K=E) would be a problem?

While the experimental evaluation demonstrates that the method works astonishingly well given the very small amount of parameters the explanation for "why this is working" is missing.

Can you provide an explanation ?

You mention that your method is similar to convolution. It would have been interesting to develop that a bit further. One could say that your shifting is strictly equivalent to a convolutional method where weights on one side are forced to zero. This resembles dilation, where the receptive field is increased without having all weights realized. This might be an interesting case for an ablation study as well.

Now what is astonishing is that computationally this works efficiently. Convolutions are highly optimized parallelized operations, but shifting groups of weights are not that well optimized. How do you manage that your method appears to work so efficiently?

If this understanding is correct, then this means that your memory requirements should be more or less equivalent with respect to storing unit activations, on the other hand you should have fewer weights to store. Depending on the size of the segments you treat within the training batches the relative memory benefit will be different.


**You wrote**: *The relative context operation in its current form basically encourages the neural network to place relevant information for the specific shifts into their corresponding channels*.

I cannot follow you here. The network is not placing anything anywhere, you are shifting the channels. If we follow your idea of the convolution, it appears to me that you increase the receptive field without using all parameters. The receptive field is not modified by the network, something outside of the receptive field remains outside. If you increase the receptive field then the network may benefit, but it cannot perform to put information into the receptive field. Could you explain your ideas when you say the model is encouraged ?

---

### Official Review · Reviewer_W1Qj · 2024-11-03

**Soundness:** 1
**Presentation:** 1
**Contribution:** 1
**Rating:** 1
**Confidence:** 2

**Summary:**

This paper claims to introduce a lightweight speech separation module. The main argument is the number of trainable parameters which is less than 500K parameters with high performance accuracy

**Strengths:**

1) The experimentation in section 1.2 on justifying the paper argument to research low footprint speech separation systems.
2) The whole topic of low footprint speech separation which an interesting machine learning problem.

**Weaknesses:**

This paper is extremely poor written. It is very difficult to read and understand the core aspects of the proposed model. Here are two confusions:

1. What is the channel size in this paper?
line 44: "In the context of this paper, we focus on single-channel source separation. Single-channel simply means that the audio was recorded using a single microphone."
which suggests this paper focuses on single channel.
And then in the introduction of "Relative context", section 3, the channel size is set to 5:
"An example of this is shown in Figure 2 where both the channel size E and the number
of subchannels K are set to 5." and Y-axis name of Figure 2.

2. What is the core concept of relative context? What is the difference between number of sub-channels (K) and channel size (E), why Figure 2 is showing channel size 5 and do not show sub-channel ?

**Questions:**

Answering my two questions under weakness will be helpful.

---

### Official Review · Reviewer_Uznw · 2024-11-07

**Soundness:** 3
**Presentation:** 3
**Contribution:** 3
**Rating:** 5
**Confidence:** 4

**Summary:**

This work introduces the concept of relative context operation, which computes the difference between an element of a sequence and its neighbors. The authors demonstrate that the relative context operation, combined with a modification of the transformer architecture referred to as Miniformer, achieves comparable performance to state-of-the-art models with higher computational efficiency.

**Strengths:**

The RCSep model has a significantly lower number of parameters, which in turn reduces computation time and memory requirements compared to other state-of-the-art models.

**Weaknesses:**

- ### Influence of Parameter K in TCRN
TCRN seems to be an interesting concept. It will be valuable to understand the influence of the parameter K of the relative context operations on the performance. Can you please report results for different values of K?

- ### Details of Miniformer Architecture
The details of the Miniformer architecture are not clear in Section 4.2.1. The 3rd image in Figure 3 just mentions Miniformer as a block without much detail. Please explain how Miniformers are different from Transformers using equations. Particularly, illustrate how attention is computed using only three weights.

- ### Performance on Realistic Datasets
The performance of the model is tested on a non-realistic dataset without including reverberations. The improvement in speech separation algorithms as demonstrated in simulated unrealistic datasets does not translate to the same level of improvements in real datasets. It would help if you show results on datasets such as LibriCSS.

**Questions:**

To enhance clarity for the readers, it would be beneficial to include the architecture of TF-GridNet with TCRN in the paper.

---

### Official Review · Reviewer_VbKV · 2024-11-07

**Soundness:** 3
**Presentation:** 3
**Contribution:** 3
**Rating:** 6
**Confidence:** 5

**Summary:**

The paper introduces a neural network module called "relative context" and uses it inside a model architecture to perform speech separation.

The use of this no parameter module enables the network to use neighboring context cheaply and the network ends up using much less number of parameters. The network also uses less compute and memory during training and inference as compared to state of the art methods.

The proposed network is compared with Sepformer, TF-GridNet, MossFormer2, TDANet, S4M-tiny in standard speech separation tasks of WSJ0-2mix and WHAM! The model is not better than more sophisticated and complex TF-GridNet and MossFormer2 but comparable or better than similar size alternatives.

The introduced module is a fixed coefficient convolutional module (with coefficients of +1 and -1 and 0) so that the gradient ends up flowing well through it. The module can replace any sequential module (like BLSTM). By replacing BLSTMs in TF-GridNet, further improvement was observed.

A negative aspect of the paper is that the model still uses a very small frame length and frame step (of 4 and 2 samples) and that it was only tested with 8 kHz artificially mixed data. I think we need to move to more common 16 kHz and more realistic datasets and move away from these 8 kHz datasets which are getting really old.

**Strengths:**

The proposed network is compared with Sepformer, TF-GridNet, MossFormer2, TDANet, S4M-tiny in standard speech separation tasks of WSJ0-2mix and WHAM! The model is not better than more sophisticated and complex TF-GridNet and MossFormer2 but comparable or better than similar size alternatives.

The introduced module is a fixed coefficient convolutional module (with coefficients of +1 and -1 and 0) so that the gradient ends up flowing well through it. The module can replace any sequential module (like BLSTM). By replacing BLSTMs in TF-GridNet, further improvement was observed.

**Weaknesses:**

A negative aspect of the paper is that the model still uses a very small frame length and frame step (of 4 and 2 samples) and that it was only tested with 8 kHz artificially mixed data. I think we need to move to more common 16 kHz and more realistic datasets and move away from these 8 kHz datasets which are getting really old. It would help the paper if the compared datasets are increased.

**Questions:**

1. Are the time-shifts always circular shifts? Why?
2. When using 7 or 3 sub-channels in time and frequency submodules, what are the time-delays used for each subchannel, please state them explicitly in the paper.
3. In 4.1.1, it is mentioned that the sequence length is halved. It is actually M/2 times reduction in length, so please correct. It is mentioned that M=4, so indeed it ends up being halved, but at this point we do not know that M=4.
4.  In 4.1.3, it seems the channel size should be changed to C. E_T (not M. E_T) and then another reduction to C.M is needed.
5. What version of SI-SDR loss is used? Is it scaling the estimate version (like in TF-GridNet paper) or not?
6. Figure 4 and 5 compute time and memory are measured on what devices? Can you also provide flops?

**Details Of Ethics Concerns:**

N/A.

---

### Meta-Review · Area_Chair_pE7F · 2024-12-16

**Metareview:**

The paper presents a novel sequence modeling module, the relative context, which requires no parameters for speech separation tasks. As a result, the whole separation network has significantly fewer parameters, providing computational and memory advantages. The proposed method is comparable to established methods such as Sepformer and TF-GridNet on standard speech separation tasks like WSJ0-2mix and WHAM!

However, there are several issues with this submission.
First of all, the proposed methods are evaluated with artificial tasks. Both WSJ0-2mix and WHAM! were developed more than 5 years ago. The speech signal is clean without background noise. This should be considered as an easier task for separation. The authors should evaluate more realistic and recent tasks such as CHiME 5+ or LibriCSS etc. Otherwise, the conclusions cannot be generalized.
There are also multiple technical errors and wrong claim, as pointed out by Reviewer ei5R.
The paper also misses lots of implementation details. The reviewers have raised multiple questions. However, the authors didn’t utilize the chance of rebuttal to answer these questions.

**Additional Comments On Reviewer Discussion:**

There is no author rebuttal.

---

### Decision · Program_Chairs · 2025-01-22

Reject